# PSMA-Directed Theranostics in Prostate Cancer

**DOI:** 10.3390/biomedicines13081837

**Published:** 2025-07-28

**Authors:** Salman Ayub Jajja, Nandini Sodhi, Ephraim E. Parent, Parminder Singh

**Affiliations:** 1NYMC-Landmark Medical Center, Woonsocket, RI 02895, USA; drsalmanjajja@gmail.com; 2University of Arizona College of Medicine, Tucson, AZ 85724, USA; nandinisodhi@arizona.edu; 3Division of Nuclear Medicine, Department of Radiology, Mayo Clinic, Jacksonville, FL 32224, USA; parent.ephraim@mayo.edu; 4Division of Hematology-Oncology, Department of Internal Medicine, Mayo Clinic, Phoenix, AZ 85054, USA

**Keywords:** PSMA theranostics, prostate cancer therapeutics, radioligand therapy, immunotherapy in prostate cancer, antibody drug conjugates

## Abstract

Following lung cancer, prostate cancer is the leading cause of cancer death in men. High-risk localized tumor burden or metastatic disease often progresses, refractory to initial treatment regimens. There is ongoing development of technology to appropriately identify high-risk patients, stage them correctly, and offer appropriate treatments to obtain the best clinical outcomes. Prostate cancer-specific membrane antigen (PSMA) is a transmembrane glutamate carboxypeptidase, which helps regulate folate absorption, and its overexpression is pathologically directly proportional and associated with prostate cancer. Increased PSMA expression is a known independent risk factor for poorer survival, and most metastatic lesions in CRPC are PSMA positive. Over the last decade, several PSMA-based PET radiopharmaceuticals have demonstrated superior sensitivities and specificities compared to traditional imaging methods. These outcomes have been demonstrated by several large clinical trials. As the data emerges, these diagnostics are being integrated into standard of care protocol to facilitate nuanced identification of malignant lesions. PSMA is also being targeted through several therapeutics, including radioligands and immunotherapies such as CAR-T, BiTEs, and ADCs. This review will discuss the landscape of PSMA-based theranostics in the context of prostate cancer.

## 1. Introduction

Prostate cancer (PCa) is the most common cancer in men, and the second leading cause of cancer death in men following lung cancer [1]. Patients with high-risk localized disease or metastatic PCa may eventually progress and succumb to their cancer. Approximately 10–20% of PCa cases progress to castration-resistant prostate cancer (CRPC) after initial androgen deprivation therapy (ADT) [2]. CRPC is a lethal form of PCa. It is critical that we develop technology to appropriately identify high-risk patients, stage them correctly, and offer appropriate treatments to obtain the best clinical outcomes. PCa-specific membrane antigen (PSMA) is a transmembrane glutamate carboxypeptidase expressed by tumor cells [3]. A representation of the structure is below (Figure 1). PSMA, encoded by the folate hydrolase 1 gene (FOLH1), helps regulate folate absorption, and its overexpression is pathologically associated with PCa [4]. Increased PSMA expression is a known independent risk factor for poorer survival, and most metastatic lesions in CRPC are PSMA positive [5,6]. Notably, neuroendocrine (small cell) PCa is notoriously PSMA negative, although most high-grade PCa exhibits PSMA directly proportional to the grade of the malignancy [5].

PSMA was identified in the early 1990s by W.G. Heston. Initially, Dr. Heston identified it through the development of a monoclonal antibody (7E11-C5.3, Capromab). Dr. Heston’s work focused on elucidating structure, molecular cloning, and functional characteristics at the Cleveland Clinic. PSMA was successfully cloned in 1993, revealing its function as a type II transmembrane glycoprotein. His colleague, Dr. Neil H. Bander, developed Capromab in conjunction [8]. In the late 1990s, Capromab Pendetide was the first drug targeting PSMA in metastatic disease. As a monoclonal antibody, it targeted an intracellular epitope of PSMA. Capromab Pendetide was approved for imaging metastatic disease when bound to Indium-111 because, at the time, there were no other prostate-specific imaging modalities. This was especially the case for detecting micro metastatic or recurrence in the setting of rising PSA. However, it had poor sensitivity, specificity, and limited utility in viable tumors because the intracellular epitope was only accessible to the antibody in dying cells [9]. This limitation fueled the development of the next generation of PSMA targeting molecules, like J591 monoclonal antibody, targeting an external domain of PSMA [10].

In the early 2010s, small-molecule PSMA ligands were advanced, which transformed PSMA PET imaging and led to its widespread adoption. PSMA-based PET radiopharmaceuticals could detect more metastases at lower radiation doses and fewer equivocal findings than conventional CT scans [11]. The proPSMA study performed in 2020 was the pivotal study that compared conventional imaging with PSMA PET-CT, leading to widespread adoption [12].

This article will present a summary of the latest developments in the field of PSMA-based diagnostics and therapeutics. The Section 1 will elaborate on the development of current PSMA-based scans, differences among various radiotracers, and current approved indications. We will also briefly discuss the interpretation criteria and clinical utility of these scans. The Section 2 will delve deeper into the development of PSMA-based therapeutics not only as radioligand therapy but also as a potential target for antibody drug conjugates and newer immunotherapy-based strategies. The review will summarize the ongoing clinical trials of these molecules and the early results that have been presented in meetings or press releases.

## 2. PSMA Directed Diagnostic Advancements in Prostate Cancer

### 2.1. Antibody-Based Agents

The first PSMA-based tracer for diagnostics was Capromab Pendetide (ProstaScint). It was approved by the FDA in 1996 specifically for biopsy-proven PCa with high risk for pelvic lymph node metastases and for detecting recurrence in patients with rising prostate-specific antigen (PSA) levels following prostatectomy. It was a monoclonal antibody labeled with In-111 that, unknown at the time, binds to an intracellular epitope of PSMA, limiting uptake at sites of viable disease. Although there was no large phase III study, ProstaScint’s approval was backed by several phase II style trials [13]. However, it was quickly supplanted by other PSMA-based tracers due to its limited sensitivity. Since it only targeted an intracellular epitope, its utility was limited to dying or dead cells.

The future of antibody-based agents is looking at conjunction with nanoparticles to detect PSMA positive PCa. There have been several studies in the last decade exploring superparamagnetic iron oxide nanoparticles (SPIONs), conjugated to PSMA-binding antibodies, that have shown promise in being used as a nanoprobe to visualize PSMA-positive lesions [14,15,16].

### 2.2. Small Molecule Targeting PSMA

Low molecular weight peptidomimetics were developed around a urea-based modality, which had very high specificity and affinity to their target PSMA. To localize tumor burden, these ligands were then tagged with radioactive isotopes. The design of these small-molecule ligands targeted the extracellular domain of the tumor cell with faster clearance from the blood and other tissues, leading to increased scan sensitivity, specificity, and ease of the cancer detection compared to ProstaScint which cleared from the body and liver very slowly, causing a delay of up to 7 days after administration of the tracer.

#### 2.2.1. Ga-68 PSMA-11

Ga-68 labeled PSMA-11 was one of the first low molecular weight radioligands developed and approved for PCa with very high sensitivity and specificity. The tracer could detect cancer at a PSA level of 2.0 with very high accuracy. Ga-68 PSMA 11 (also known as ^68^Ga-PSMA-HBED-CC, Illuccix or Locametz), which was studied at the University of Heidelberg in the early 2010s. Ga-68 PSMA-11 has been approved in the US as a PSMA diagnostic since 2020, given its high detection rates at very low PSA levels. A phase III diagnostic efficacy trial found the sensitivity and specificity of ^68^Ga-PSMA-11 PET to be 0.40 and 0.95, respectively. The results of ^68^Ga-PSMA-11 PET were positive in 40 of 277 (14%) patients for pelvic nodal metastasis. The true positives in the trial were 75 of 277 patients (27%), resulting in sensitivity, specificity, positive predictive value, and negative predictive value to be 0.40 (95% CI, 0.34–0.46), 0.95 (95% CI, 0.92–0.97), 0.75 (95% CI, 0.70–0.80), and 0.81 (95% CI, 0.76–0.85), respectively, for pelvic nodal metastasis [17]. The scan is approved for patients with suspected metastasis who are candidates for initial definitive therapy or with suspected recurrence based on an elevated serum PSA level [18].

#### 2.2.2. Ga-68 Gozetotide

In March 2025, the FDA approved a novel solid target cyclotron production of Ga-68 formulation of ^68^Ga-PSMA-11, Gozetotide (GOZELLIX). GOZELLIX offers a significant logistical advantage with an extended post-radiolabeling shelf life of up to six hours—markedly longer than prior formulations of ^68^Ga-PSMA-11. As a result, it enables broader geographic distribution and more flexible imaging schedules. Its approval was supported by two pivotal clinical trials, PSMA-PreRP and PSMA-BCR, which demonstrated a specificity of 90% and a true positive rate of 91% [17,19].

#### 2.2.3. ^18^F-DCFPyL (Pylarify)

^18^F-DCFPyL (Pylarify) is a next-generation fluorine-18 labeled small molecule targeting PSMA. This was investigated in the OSPREY trial, a prospective, multicenter, phase III trial published in 2021. It has a higher sensitivity and specificity for detecting lymph node and distant metastases compared to a ^68^Ga-PSMA-11, validating its use for initial staging [20]. The study’s initial staging sensitivity of ^18^F-DCFPyL was 40.3% (95% CI: 30.8% to 50.6%), specificity was 97.9% (95% CI: 94.0% to 99.6%), and the positive predictive value was 86.7%. In the cohort looking at biochemical recurrence, the data demonstrated clear utility to localize sites of recurrence that are often missed on CT. The detection rates increased with higher PSA levels, reaching above 90% in patients with PSA levels over 2 ng/mL. The approval was largely backed by the CONDOR phase III trial, which found ^18^F-DCFPyL localized recurrence in 60–70% of patients, even with PSA < 0.5 ng/mL [21].

One principal advantage of ^18^F-DCFPyL over ^68^Ga-PSMA-11 was the longer half-life of F-18, allowing for centralized production, and lower positron energy, allowing for better image resolution, leading to high sensitivity and specificity for small lymph node and bone metastases. The indications approved for ^18^F-DCFPyL are like ^68^Ga-PSMA-11. ^18^F-PSMA-1007, a similar tracer in the pipeline that was developed alongside Pylarify, is not approved in the United States but is widely used in Europe and Australia. There are two significant ongoing phase 3 trials looking at its diagnostic performance [22,23].

#### 2.2.4. ^18^F-rhPSMA-7.3

In 2023, the FDA approved ^18^F-rhPSMA-7.3 (also known as Posluma or flotufolastat F-18). This molecule is a “radio-hybrid” PSMA that can be theoretically labeled with both diagnostic and therapeutic isotopes. ^18^F-rhPSMA-7.3 also demonstrates lower urinary excretion and retention in the urinary bladder, allowing for enhanced visualization of primary disease with adequate hydration, as demonstrated in Figure 2 below.

Similarly, the indications of ^18^F-rhPSMA-7.3 are patients with suspected metastasis eligible for definitive therapy and patients with rising PSA after prior treatment [24]. The LIGHTHOUSE and SPOTLIGHT trials were key to its approval [25,26]. SPOTLIGHT showed detection rates of 60–70% even at low PSA (<1 ng/mL), and. LIGHTHOUSE confirmed these findings by reporting a sensitivity of 60% for pelvic lymph node metastases and a specificity of 96%.

A summary of the approved PET tracers is below in Table 1.

### 2.3. Future of PSMA Directed Diagnostics

Additional upcoming trials include COBRA [27] and CLARIFY [28] which are looking at copper-64–labeled bispecific PSMA PET tracer (^64^Cu-SAR-bisPSMA). The advantage of the ^64^Cu-SAR-bisPSMA may be increased sensitivity and specificity. ^64^Cu-SAR-bisPSMA has a longer half-life, allowing for delayed imaging and potentially greater sensitivity. So far, ^64^Cu-SAR-bisPSMA has demonstrated safety and efficacy in detecting PSMA-expressing lesions in phase I trials [29].

Another tracer being explored is Fluorescent PSMA-Targeted Dye through the ProMOTE Study [30]. The application of this tracer will be intraoperative visualization of the tumor burden. The early-stage trials in Europe have shown promise in improving surgical outcomes [31]. Similarly, in the Netherlands, there was a phase I/II study in 2024 looking at ^111^In-PSMA radio-guided surgery. The participants were monitored by ^18^F-PSMA PET/CT to analyze results. This small study found that using a radioactive PSMA tracer both prior to and intraoperatively may improve the identification and removal of malignant lymph nodes [32].

Comparably, there was another small study of thirty patients in Italy utilizing ^99m^Tc-PSMA intra-operative lymph node imaging and surgery. The study demonstrated that using a target-to-background ratio ≥ 2 to identify suspicious nodes could spare >50% extended pelvic lymph node dissection and would identify additional pN1 patients compared to PSMA PET [33]. A comparison of ongoing PSMA tracer trials is summarized in Table 2 below.

### 2.4. Limitations

It is increasingly evident that PSMA-targeted tracers offer superior lesion-specific insights into disease biology compared to PSA kinetics and conventional imaging modalities. The NCCN, as of 2022, recommends a PSMA PET for high-risk newly diagnosed and biochemical recurrent disease evaluations [34]. However, like all diagnostics, there are limitations of PSMA-based tracers.

One of the limitations of PSMA tracer-based scans is that traditional CT and MDP bone scintigraphy and bone scans for treatment monitoring are still cheaper and more widely available at this time, despite PSMA-based scans being more sensitive and specific. Although centers in lower-resource settings may adapt these technologies using emerging lower-cost tracers or mobile cyclotron solutions, potentially improving global accessibility. ^99m^Tc-PSMA offers a lower-cost, more accessible alternative to PSMA-PET, making it a viable solution for centers lacking PET/cyclotron infrastructure [35]. Currently, PSMA-based imaging is the emerging gold standard for both diagnostics and treatment response monitoring in Germany, where VISION, TheraP, and LuPSMA trials were led [36].

Another limitation of PSMA-based tracers is that PSMA expression is very heterogenous. There is expression in the tissue of salivary glands, kidneys, lacrimal glands, and the small intestine [37]. To combat the heterogenous expression, there is an ongoing search for more specific molecules such as ACP3 Ga based tracers, radio conjugates consisting of a gastrin-releasing peptide receptor (GRPR) antagonist (i.e., ^68^Ga-NeoB). ACP3 is a promising alternative targeting prostatic acid phosphatase (PAP), an enzyme primarily produced by the prostate gland. PAP has higher expression in PCa tissues and minimal presence in normal organs, reducing off-target effects and improving imaging specificity [38]. There is also ongoing research developing gastrin-releasing peptide receptor (GRPR) antagonists, given its expression in early-stage hormone-sensitive cancers, unlike PSMA, which is more present in late-stage cancer [39,40].

Given these advantages, the alternative tracers discussed above are being developed complementary to PSMA-based theranostics to be used when patients have low PSMA markers [41,42,43,44]. Choline tracers were previously approved for PET scans in biochemical recurrent PCa; however, PSMA scans have now demonstrated an improved detection rate as compared to choline PET scan. Current research is exploring choline tracers as adjuncts to PSMA-based theranostics. Another tracer is FACBC, a radiolabeled amino acid, which was developed with non-urinary excretion reducing background signal from the bladder and a fast imaging window. Yet, they also have lower sensitivity than PSMA PET/CT, especially in detecting small metastatic lymph nodes [45].

### 2.5. Clinical Applications of PSMA PET Scan in Prostate Cancer

For PSMA’s diagnostic application, trials such as the Avidity trial in the UK are trying to expand its indications into aggressive prostate cancers and determine if PSMA diagnostics can increase accuracy and personalized treatment plans for patients. However, there remains an unmet need to define and standardize two critical roles for which PSMA PET imaging is being utilized or validated in clinical practice.

#### 2.5.1. Prognostic Utility and Role in Treatment Response Assessment

Firstly, as a prognostic tool, PSMA PET is being applied across all stages of PCa—ranging from initial diagnosis and staging to biochemical recurrence, metastatic hormone-sensitive disease, and CRPC. During clinical practice, we have noticed PSMA PET Scans perform better at visualizing the disease.

Figure 3 shows a PSMA PET scan of a 74-year-old man with prostate cancer, Gleason grade 7, status post prostatectomy and salvage external beam radiation therapy. He presented with rising PSA consistent and most recent PSA of 1.0 ng/mL. Targeted external beam radiotherapy to the lymph node was subsequently performed, resulting in a drop of PSA to <0.10 ng/mL. The patient had normal conventional imaging scans, which showed no evidence of disease. However, a PSMA PET scan ordered due to clinical suspicion showed evidence for metastatic lymph node-positive disease. The patient elected to proceed with radical prostatectomy, and the biopsy findings aligned with the PSMA PET scan evidence of nodal metastases.

Its prognostic value is being assessed both independently and in conjunction with established risk stratification frameworks such as the NCCN and EAU guidelines [34,46].

Secondly, PSMA PET is being explored for response assessment in patients undergoing therapy. Emerging protocols now leverage PSMA-based tracers to monitor treatment response, expanding their utility beyond diagnosis alone. A crucial next step in advancing this approach is the development of a validated framework for response evaluation. A sample comparison of responses is demonstrated in Figure 4 below. This would require standardized methods for capturing tracer-specific features from PET scans and integrating them into consistent criteria for treatment response assessment. Since PSMA tracers provide superior lesion-specific insight, more accurate than PSA trends, they are becoming integral to various response evaluation criteria such as the PROMISE V2 [47], RECIP 1.0 [48], and PPP Criteria [49].

#### 2.5.2. Framework for Standardizing PSMA PET Response Assessment

The PROMISE V2 (Prostate Cancer Molecular Imaging Standardized Evaluation) framework aims to standardize the reporting of PSMA PET findings across staging, restaging, and therapy response assessments. It enables consistent and accurate characterization of PSMA expression, which has been shown to correlate with patient survival [50]. The RECIP 1.0 (Response Evaluation Criteria in PSMA PET/CT) protocol combines assessment of both total tumor volume and the appearance of new lesions to evaluate treatment response [48]. Given its focus on total tumor burden, RECIP 1.0 is particularly well-suited for patients with extensive disease. The PPP criteria (PSMA PET Progression Criteria) have been widely used in clinical trials investigating PSMA-targeted therapies but have not yet been integrated into routine clinical practice [40,41]. This framework incorporates clinical parameters, laboratory values, and PSMA PET imaging findings to define disease progression, drawing from the structure of the previously established Prostate Cancer Working Group 2 (PCWG2) criteria. Unlike RECIP 1.0, which assesses the total tumor burden, PPP criteria emphasize changes in a dominant lesion and the emergence of new lesions, making them more suitable for patients with limited disease burden. Table 3 below compares the RECIST, RECIP, and PPP2 protocols.

A recent study showed that PROMISE and PPP2 provide a reproducible global standard for interpreting PSMA PET. In this study, the C-index for predictive accuracy was 0.8 for both the visual and quantitative measures. Both significantly outperform the NCCN and EAU clinical risk models [50]. The structured data captured through PROMISE V2 not only supports the calculation of emerging metrics such as RECIP and PPP2 but also serves as a foundation for the development of future response assessment criteria. This protocol is now widely recognized within the nuclear medicine community and is being increasingly adopted by additional institutions and incorporated into clinical trials.

#### 2.5.3. Limitations and Evolving Challenges in PSMA PET Interpretation

One key caveat that is important to understand is that the use of these scans and responses was developed in hormone-refractory patients, and thus, further studies are necessary to validate them in hormone-sensitive settings [51]. There are several ongoing studies, such as the PSMAtrack Study (NCT06479187) and CHAMPION Study (NCT05188911), to explore this hypothesis. Additionally, emerging evidence suggests that androgen deprivation therapy (ADT) can modulate PSMA expression, leading to heterogeneous increases in tracer uptake, particularly in bone metastases. This phenomenon, often referred to as a “flare,” may mimic disease progression on imaging studies [52]. Secondly, it is known that PSMA expression in newly diagnosed patients can be augmented by androgen blockade. Augmentation suggests that this need is clearly defined to avoid calling it progression, and or differentiate from possible flare observed on nuclear medicine bone scan. PSMA expression may decrease with potent androgen deprivation or other aggressive treatments [53]. Congruence of the scan to treatment is directly impacted by the type of treatment. There is high congruence with radioligand therapy (i.e., Pluvicto) [54], moderate-high congruence with AR-targeted therapy [53,55], and moderate-high congruence with chemotherapy [56]. This is because, in the short term, it increases uptake, as PSMA synthesis via FOLH1 is inversely correlated to AR activation. However, in the long term, the antiproliferative effects of ADT outweigh the PSMA expression [57]. Therefore, while PSMA PET imaging offers valuable insights into disease status, its interpretation, especially in the early phases of ADT, should be contextualized within the broader clinical picture. Biochemical markers and clinical assessment to guide accurate treatment decisions should be incorporated. There are no current standards of care recommendations for the timing of PSMA PET regarding ADT. Clinically, many patients undergo imaging while on current therapies without trying to time PET for maximum uptake.

### 2.6. Conclusions

In summary, over the last decade, several PSMA-based diagnostics have demonstrated superior sensitivities and specificities compared to traditional imaging methods. These outcomes have been demonstrated by several large clinical trials. As the data emerges, these diagnostics are being integrated into standard of care protocol to facilitate nuanced identification of malignant lesions. PSMA is also being targeted through several therapeutics, described in the next section.

## 3. PSMA Directed Therapy Advancements in Prostate Cancer

Metastatic castration-resistant prostate cancer (mCRPC) remains an incurable disease and represents the most lethal phase of PCa, accounting for the majority of PCa-related mortality. Chemotherapy, immunotherapy, and bone-targeting radioactive isotopes are approved as subsequent lines of therapy. PSMA-based radioligand therapy has now emerged as an option for newly evolved mCRPC, and after progression on chemotherapy. This section will talk about PSMA-based radioligand therapy and other emerging treatments that are in clinical trials and have the potential to change the paradigm of PCa treatment. In the past 20 years, substantial therapeutic advancements have been made in treating patients with mCRPC. These include multiple hormonal, chemotherapeutic, immunotherapeutic as well as radiotherapeutic options. For the purpose of this review article, we shall be concentrating on the PSMA-directed therapeutic options.

### 3.1. PSMA-Based Radioligand Therapy

#### 3.1.1. ^177^Lu-PSMA-617

^177^Lu-PSMA-617 (Pluvicto) is a small molecule made up of a combination of a β-emitting lanthanide, ^177^Lu, and a pharmacophore, PSMA-617. This radioligand therapy has gained a lot of attention in recent times. One of the reasons for this was the advantageous long half-life of ^177^Lu. This allowed it to be transported to centers further away from its production site to where its therapeutic usage was required [58]. Apart from this, the property of ^177^Lu, causing intense localized cytotoxic effect, made it an avenue worth exploring to couple it with PSMA-617, which has a very strong affinity for PSMA-expressing PCa cells.

Initially approved by the FDA in March 2022, based on the results of the VISION trial [54]. Pluvicto was initially indicated for patients with PSMA-positive mCRPC previously treated with androgen receptor pathway inhibitors (ARPIs) and taxane-based chemotherapy. In March 2025, following the positive results of the PSMAfore trial, the FDA expanded its indication to include chemo-naïve patients. The trial compared ^177^Lu-PSMA vs. an androgen receptor pathway inhibitor (ARPI) in patients who had progressed on ADT and ARPI in chemo-naïve patients. Patients who received ^177^Lu-PSMA-617 had a median radiographic progression-free survival of 9.30 months as compared to 5.55 months (HR: 0.41 [0.29–0.56]) in the control, ARPI change group [59]. Not only did the patients receiving Pluvcito have a longer median radiographic progression-free survival, but they also had a better safety profile as compared to the ARPI change group. Patients in the Pluvicto arm suffered from fewer adverse events. This combination of clinically meaningful improvement and better safety profile led to the expanded indication of 177Lu-PSMA-617 in chemo-naïve patients as well.

Other trials that have contributed to advances in ^177^Lu-PSMA-617′s efficacy in mCRPC are summarized in Table 4. The recommended regimen—7.4 GBq (200 mCi) intravenously every six weeks for up to six cycles, or until disease progression or unacceptable toxicity—remained unchanged.

#### 3.1.2. Combining ^177^Lu-PSMA-617 with Novel Agents

Pluvicto is being investigated in combination with other agents like enzalutamide and pembrolizumab. The ENZA-p trial was designed for patients with no prior exposure to ARPI developing mCRPC to receive enzalutamide plus Pluvicto. The rationale was based on ARPI-induced upregulation of PSMA expression. The trial indicated deeper and more durable PSA declines and an extension of radiographic progression-free survival compared with historical ARPI monotherapy, all within a manageable safety profile [55].

This strategy is especially valuable if the patient has not received prior ARPI for their advanced disease. By recent estimates, only a third of newly diagnosed patients are receiving dual androgen blockade, but the uptake of guideline-directed therapy has been increasingly observed in the last few years [60]. A phase I trial explored the combination of a single dose of ^177^Lu-PSMA-617 followed by maintenance pembrolizumab in patients with mCRPC [61]. The rationale behind the study was that targeted radioligand therapy might boost the efficacy of immune checkpoint inhibitors by enhancing antigen priming and reprogramming the immunosuppressive tumor microenvironment to support stronger effector responses [62]. Long-term follow-up demonstrated a 5% complete response rate and a 47% confirmed partial response rate. The study also showed that a single priming dose of ^177^Lu-PSMA-617 followed by pembrolizumab maintenance was safe and exhibited promising preliminary activity in patients with mCRPC. These new combination strategies will help in incremental benefit in the clinical outcomes of our patients.

#### 3.1.3. ^177^Lu-PSMA-617 in Earlier Stages of PCa

To provide better patient care and to effectively reduce disease burden, there has been growing interest to trial these therapies in earlier stages of disease, apart from the mCRPC stage of PCa. For this purpose, in the localized stage of PCa, a small pilot study was conducted, which demonstrated that up to two cycles of neoadjuvant ^177^Lu-PSMA-617 before robotic-assisted radical prostatectomy are well-tolerated, with no significant increase in perioperative complications and promising early signals of histologic response [63]. Also, a phase III trial, PSMAdditon, is aiming to evaluate ^177^Lu-PSMA-617 in the mHSPC setting of the disease. The interim analysis for this trial is yet to be reported [64].

Additionally, a phase II trial is planned to assess this therapy in earlier stages of PCa. The PRELUDE trial will assess this agent prospectively in patients with high-risk localized PCa.

### 3.2. Novel PSMA Radioligand Therapies

Limitations of existing radioligand therapies are seen in the form of patients who are still progressing on the currently available radioligand therapies. For this purpose, in addition to beta-emitters, such as ^177^Lu-PSMA-617, there is growing international interest in alpha-emitting radioligand therapies. These agents, particularly ^225^Ac-PSMA, offer enhanced cytotoxicity and deeper tumor penetration, which may benefit patients with micrometastatic or treatment-refractory disease. ^225^Actinium has a relatively long half-life of ~10 days, and it emits alpha particles with energy ranging from 5.8 to 8.4 MeV [65]. Although ^225^Ac-PSMA had high antitumor efficacy but it is also worth noting that such alpha emitters have higher uptake in the salivary glands, leading to more severe xerostomia and dose-limiting toxicity as compared to ^177^Lu-PSMA-617 [66]. The WARMTH Act trial [67] explored ^225^Ac-PSMA as a last-line therapy in mCRPC patients who had progressed despite or were ineligible for ARPIs, taxanes, and ^177^Lu-based therapies. ^225^Ac-PSMA radioligand therapy has demonstrated robust antitumor activity in mCRPC and represents a viable option for patients who have progressed on prior approved treatments [68]. A recent systematic review [67] and meta-analysis reported a PSA50 response rate of 65% with ^225^Ac-PSMA therapy—comparable to the 66% observed with ^177^Lu-PSMA-617 in the VISION trial. These findings highlight the potential of alpha emitters and underscore the need for prospective randomized studies to validate their efficacy. These studies will also allow us to assess their efficacy in heavily pre-treated patients, with a more resistant form of disease, and if they are potentially better than ^177^Lu-PSMA-617 in such disease settings.

Furthermore, several other novel radioisotopes are under investigation for therapeutic use in mCRPC, including Thorium-227, Terbium-161, and Lead-212 (Table 5). Each offers distinct radiophysical properties that may expand the utility of PSMA-targeted radionuclide therapy and enhance therapeutic precision.

The rationale for exploring more alpha emitters like ^227^Thorium (^227^Th) is that the alpha particle emissions deliver very high linear energy transfer within a brief distance, inducing double-strand DNA breaks that efficiently kill PSMA-positive cells and metastases while sparing surrounding normal tissue [69]. Compared with ^177^Lu PSMA 617, which delivers lower linear energy transfer over extended tissue distances, ^227^Th confines its high-intensity alpha emissions to a brief path length, minimizing exposure of healthy tissue while more effectively eradicating cancer cells. ^227^ Th decays into ^223^Ra, which further decays via alpha emissions into ^219^Ra. [70] Despite its promise, ^227^Th development faces significant hurdles. Its production depends on Actinium-227, which in turn is generated through neutron irradiation of scarce Radium-226 targets. Like all alpha-emitting isotope productions, limited reactor throughput and generator availability constrain global supply, creating a bottleneck in precursor material for clinical-scale radiopharmaceutical manufacturing [71].

Similarly, ^161^Tb, which decays with a half-life of approximately 6.95 days, closely matches ^177^Lu’s half-life of 6.64 days. It undergoes β-emission decay to ^161^Dy with additional creation of low-energy Auger electrons, delivering high linear energy transfer at subcellular distances. This emission profile produces potent localized DNA damage in PSMA-expressing tumor cells and metastases, potentially improving therapeutic efficacy and minimizing toxicity to surrounding healthy tissue [72].

^212^Pb, with a convenient half-life of 10.64 h, offers a similar advantage over ^177^Lu. The key hurdle in its development revolves around the limited ^212^ Pb supply and complex radiochemical purification steps [73,74]. Its short half-life reduces prolonged unnecessary exposure to healthy tissue. Each disintegration of ^212^Pb emits an energy of 6–8 MeV compared to 0.4 MeV for each disintegration of ^177^Lu [75].

In addition to this, we have recently seen encouraging data about radio-antibody drug conjugates targeting PSMA. ProstACT SELECT Trial of TLX591 showed a rPFS of 8.8 months in 23 patients with diagnoses of progressive mCRPC. This reaffirms faith in the future and upcoming investigations to explore more efficacious treatment therapies [76].

### 3.3. PSMA-Based Immunotherapies

#### 3.3.1. Chimeric Antigen Receptor T-Cell (CAR-T) Therapy

PCa is often described as an “immune desert” due to low immunogenicity, a low number of lymphocytes that can penetrate the tumor, and an unfavorable micro-environment [77].

The PSMA-targeted CAR-T-cell therapies have shown encouraging results in early-phase clinical trials. Some of the most promising advances involve “armored” CAR constructs, which are engineered to resist the immunosuppressive conditions of the tumor microenvironment.

In this approach, a patient’s own T-cells are collected through leukapheresis and genetically modified to express CARs that specifically target PSMA, which is highly expressed on PCa cells and their metastases. This targeted approach helps overcome the immune-excluded nature of prostate tumors by enabling T-cells to effectively recognize and attack cancer cells. Importantly, CAR T-cells can recognize tumor antigens in a manner independent of MHC presentation, which contributes to their potent and sustained cytotoxic activity [78], as shown in Figure 5.

Despite its relevant recent discovery for therapy in PCa, significant advances have been made in the second generation of CAR-T, which has achieved far better results than its predecessor. The major difference between the two generations was the addition of CD28 as a costimulatory molecule in the PSMA-CAR [79]. This led to 75% of second-generation CAR-T groups achieving a complete remission as compared to ~13% in first-generation CAR-T [80]. In the fourth generation of CAR-T-cells, a suicide gene has also been introduced to provide a kill-switch when adverse toxic effects happen.

Limitations of this therapy include that CAR-T monotherapy is not as effective as promised in preclinical and animal studies and has to be combined with other therapeutic agents to provide a more favorable environment. Radiotherapy improves the local tumor environment, which provides more potent penetration and essentially better efficacy of CAR-T cell-directed therapy.

The ultimate promise of this therapy is to change the immune microenvironment of PCa into a hot tumor, allowing for improved immunogenicity and potentially curing the disease. Several trials are exploring this promising avenue of therapy (Table 6).

#### 3.3.2. Bispecific T-Cell Engager (BiTEs) Introduction and MOA

Bispecific T-cell engagers (BiTEs) are a class of engineered antibody-based immunotherapies designed to simultaneously bind the CD3 molecule on T-cells and a tumor-specific antigen, such as PSMA in PCa. This dual binding effectively bridges T-cells and cancer cells, triggering T-cell activation and subsequent tumor cell lysis (Figure 6).

A key advantage of BiTEs lies in their “off-the-shelf” usability—they are not patient-specific, which simplifies manufacturing, enhances accessibility, and improves storage and administration logistics.

BiTE therapy has evolved significantly since its early iterations. The first agent, Pasotuxizumab (AMG 212), a bispecific anti-CD3/anti-PSMA protein, demonstrated limited efficacy due to the development of treatment-emergent anti-drug antibodies (TE-ADAs) following subcutaneous administration. While Acapatamab overcame this immunogenicity challenge, its short half-life necessitated continuous infusion to maintain therapeutic levels, limiting its practicality in clinical settings.

More recently, IgG-based BiTE constructs have been developed with extended serum half-lives, offering sustained drug exposure and improved feasibility for real-world use [81].

#### 3.3.3. Limitations of BiTEs

A disadvantage of BiTEs is that the cytokine release syndrome (CRS) and poor pharmacokinetic profile significantly limit the potential of this drug class. CRS arises when engager-mediated T-cell activation unleashes a burst of inflammatory cytokines, most notably IL-6, TNF-α, and IFN-γ. Those initial signals draw in monocytes and macrophages, which then amplify the inflammatory cascade. BiTEs engineered for very high

CD3 affinity can exaggerate this feedback loop, driving severe CRS that has forced the early termination of multiple clinical trials.

#### 3.3.4. Future of BiTEs

Next-generation BiTEs are being engineered to mitigate the risk of cytokine release syndrome (CRS) by modulating their affinity for CD3, thereby potentially disrupting the feedback loop that drives excessive T-cell activation. One such agent, JANX007, a novel PSMA-targeted Tumor-Activated T-cell Engager (TRACTr), has demonstrated highly promising early results. This molecule significantly reduces off-tumor T-cell activation by incorporating a CD3-binding domain with more than 600-fold lower affinity for human CD3, limiting systemic immune activation [82]. Similarly, AMG 340 is another innovative molecule designed with a built-in safety mechanism. It employs a low-affinity anti-CD3 arm to attenuate the strength of T-cell engagement, thereby reducing the risk of CRS—the most common and serious adverse event associated with BiTE therapy. While its clinical efficacy in PCa remains modest, AMG 340 serves as a compelling proof of concept for improving the safety profile of bispecific T-cell engagers in solid tumors [83]. Table 7 provides a summary of the trials exploring the efficacy and safety of BiTes.

#### 3.3.5. Antibody–Drug Conjugates Introduction and MOA

In parallel, PSMA-directed antibody–drug conjugates (ADCs) are also under active investigation. These agents target antigens like PSMA to ferry cytotoxic therapies straight into cancer cells. Some of these ADCs are summarized in Table 8. The antibody moiety recognizes and binds with the antigen of interest, e.g., PSMA, that is preferentially overexpressed in tumor cells. Following the binding, receptor-mediated internalization occurs, which leads to the payload delivery from the linked drug and ultimately the cytotoxic effect (Figure 7).

#### 3.3.6. Limitations of Antibody–Drug Conjugates

A total of 13 ADCs have gained FDA approval in other malignancies; however, no ADC has been approved yet for PCa. Possible reasons for this are that PSMA is also expressed in normal tissue, like vascular tissue, salivary glands, and kidneys. This leads to dose-limiting toxicities before delivering an effective anti-tumor effect [84]. NCT02991911, which investigated the use of ADC MEDI3726 in mCRPC patients, found that the toxicity profile was unacceptable. Toxicities most notably included capillary leak syndrome, hypoalbuminemia, and edema. These adverse events ultimately outweighed the therapeutic benefits, leading to the termination of the clinical trial.

Apart from this, the intrinsic affinity of PCa to metastasize to bones is also a limiting factor for this therapy. As bone metastases are relatively poorly vascularized, which leads to impaired payload delivery [85]. In contrast, PSMA-targeted radioligands are low molecular weight, as compared to high molecular weight ADCs, compounds that rapidly extravasate through the fenestrated sinusoidal capillaries of red-marrow–rich trabecular bone, diffuse uniformly within metastatic niches, and bind PSMA with high affinity to trigger receptor-mediated internalization (up to 75% within hours), thereby concentrating β- or α-emitters intracellularly to deliver potent, localized high-LET radiation to bone lesions while sparing surrounding healthy tissue.

#### 3.3.7. Future of Antibody–Drug Conjugates

There is a lot of promise in this regard with agents like ARX517 [86] demonstrating good antitumor activity and tolerable safety profiles in early clinical trials [84]. Recently, a new molecule, FOR46, was tested in the mCRPC population [87]. In this phase I first-in-human study, a composite response rate of 32.5% was seen, and a PSA50 response of 36% was confirmed in an evaluable set of 39 patients. The treatment discontinuation rate due to toxicity was very low at 9% suggesting an encouraging benefit-to-risk ratio. Another phase I study aimed to evaluate the safety and efficacy of a dual variable ADC in patients with mCRPC. In which ABBV-969, a dual variable domain IgG1 drug conjugate comprising PSMA and STEAP1 binding domains and a topoisomerase 1 inhibitor (Top1i) payload, is being evaluated [88].

Building upon the earlier discussion, Table 9 outlines the available data on clinical responses and safety profiles of some of the PSMA-targeted CAR-T-cells, BiTEs, and ADCs [81,89,90,91,92,93].

Collectively, these diverse and rapidly evolving platforms underscore the expanding role of PSMA as a versatile and clinically actionable target. Continued investigation through well-designed clinical trials will be critical to determine optimal sequencing, combination strategies, and positioning of these therapies within the broader mCRPC treatment landscape.

### 3.4. Other Molecules

While small-molecule ligands targeting PSMA have been effectively used to deliver radionuclides, such as in ^177^Lu PSMA 617, no small-molecule non-radiolabeled drug conjugates directed at PSMA have received regulatory approval to date. A new study recently investigated a new class of PSMA-targeted small-molecule drug conjugates by coupling the targeting unit of Pluvicto, termed OncoPSMA, to cytotoxic auristatin payloads using cleavable linkers including valine citrulline, disulfide bridges, and a postprolyl peptidase cleavable linker called Gly Pro. Among these, the Gly Pro linker demonstrated the most efficient and tumor-selective payload release based on mass spectrometry biodistribution studies in tumor-bearing mice. The mechanism involves PSMA-mediated internalization and enzymatic cleavage by intracellular or tumor-associated proteases such as fibroblast activation protein. In therapeutic studies, OncoPSMA conjugates with Monomethyl auristatin E or Monomethyl auristatin F showed strong antitumor effects, and when combined with a tumor-targeted interleukin 2 fusion protein, induced complete and durable remissions in all animals. Although single-agent therapy did not result in lasting cures and may depend on tumor microenvironment factors, these small-molecule drug conjugates (SMDCs) offer a promising and scalable therapeutic approach for patients with advanced PCa who may not respond to existing radioligand therapies [94].

## 4. Conclusions

PSMA-guided theranostics represents a rapidly evolving field with numerous promising agents under investigation. This approach has the potential to significantly improve outcomes for patients with PCa by enabling more precise detection and targeted treatment of the disease. PSMA-based diagnostics clearly offer superior sensitivity and specificity compared to traditional cross-sectional imaging, allowing for more accurate identification of metastatic or recurrent disease, which can lead to timely and more effective clinical decision making. As physicians gain experience with image interpretation, PSMA’s role in cancer diagnosis and management is expected to expand further. Likewise, therapeutic targeting of PSMA is opening new avenues for treatment, including novel modalities and applications in earlier stages of disease, potentially improving survival rates and quality of life. Overall, the integration of PSMA-guided diagnostics and therapeutics offers significant advantages in personalizing care and optimizing treatment strategies for patients with CRPC. Figure 8 summarizes the current clinical applications, near-term innovations, and future directions of PSMA-targeted imaging and therapies, including radioligands and emerging immune-based strategies.

## Figures and Tables

**Figure 1 biomedicines-13-01837-f001:**
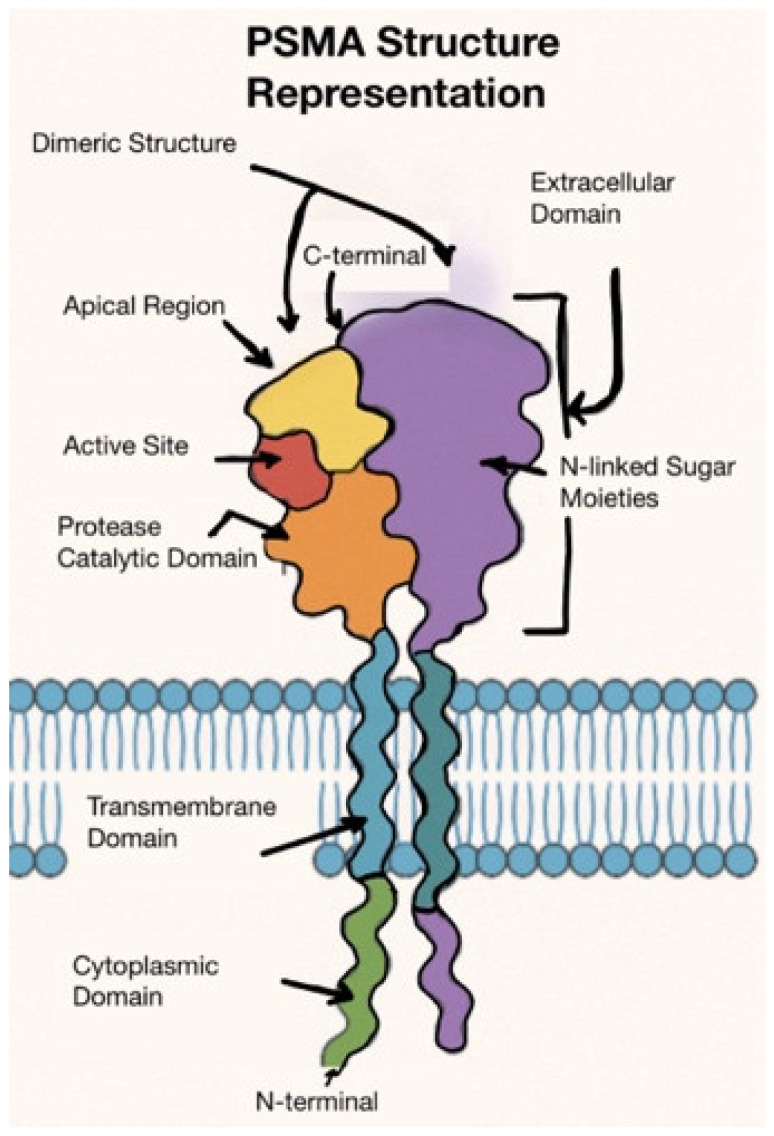
Cartoon representation of PSMA’s structure [7].

**Figure 2 biomedicines-13-01837-f002:**
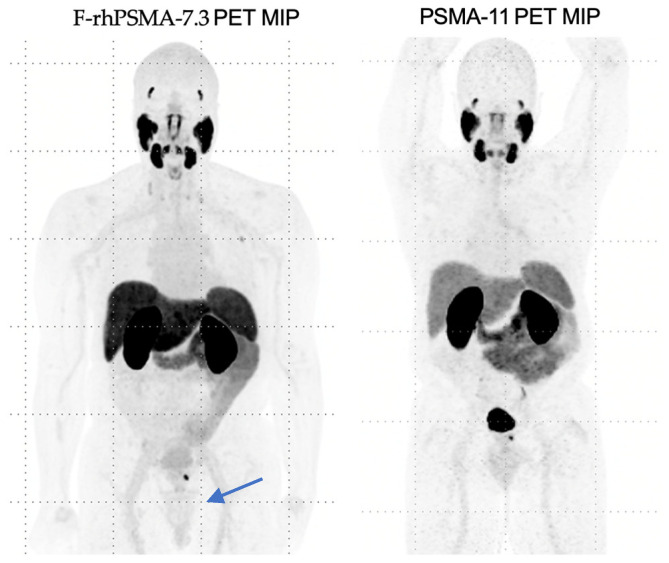
Comparison of ^18^F-rhPSMA-7.3 imaging to PSMA-11, highlighting limited bladder uptake by ^18^F-rhPSMA-7.3.

**Figure 3 biomedicines-13-01837-f003:**
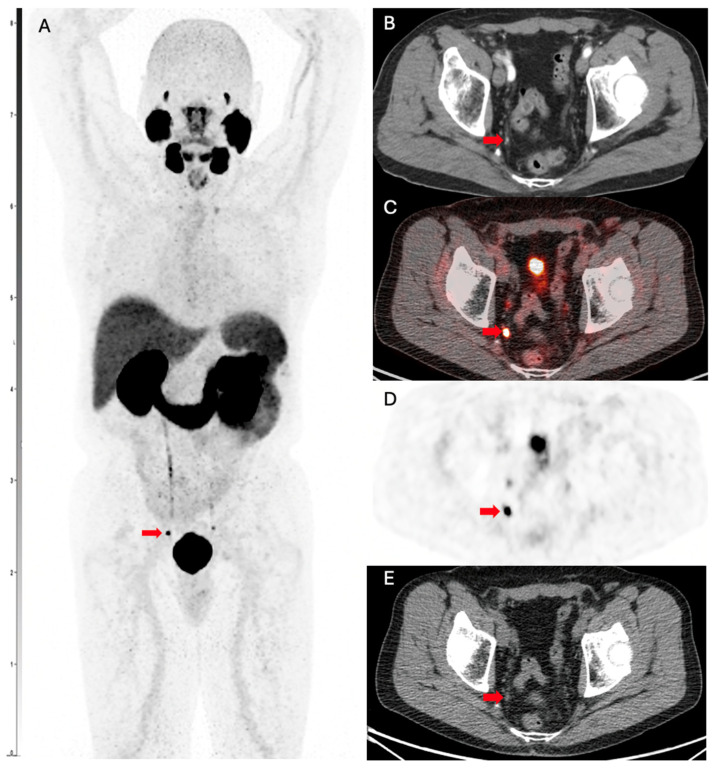
PSMA PET scan showing evidence for metastatic lymph node disease in a patient with PCa. (**A**) Ga-68 PMSA-11 PSMA/CT whole body maximum intensity image (MIP) shows a focus of nodular uptake (red arrow) adjacent to the right ureter. (**B**) Transaxial contrast-enhanced CT demonstrates a 0.3 cm right internal lymph node (red arrow), which was not mentioned on the clinical report. Transaxial slices of the Ga-68 PMSA-11 PSMA/CT show intense PSMA-11 accumulation in the lymph node (red arrow) on fused (**C**) PET/CT, (**D**) PET only, and (**E**) CT only.

**Figure 4 biomedicines-13-01837-f004:**
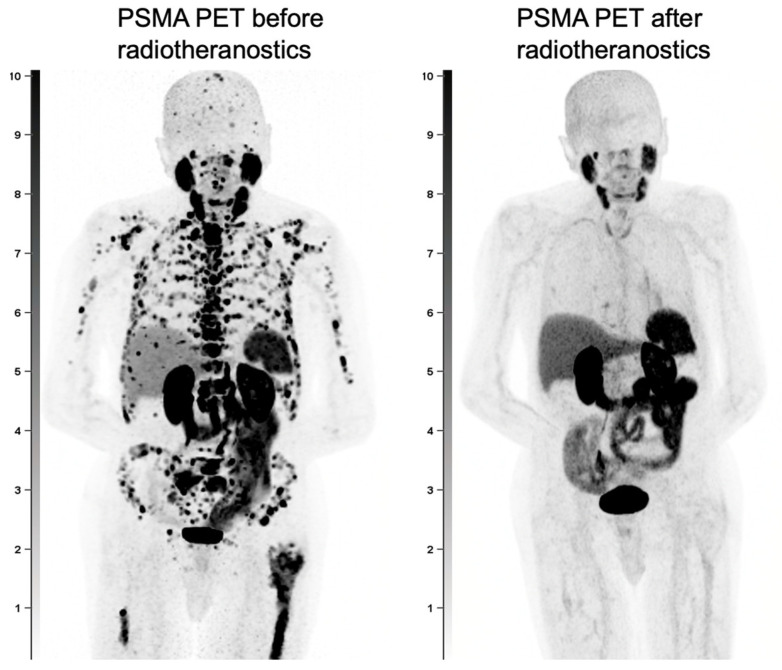
A male with PCa with extensive metastatic disease throughout the skeleton and lymph nodes, seen on the pre-radiotheranostics PET and with near complete resolution of disease after radiotheranostic treatments.

**Figure 5 biomedicines-13-01837-f005:**
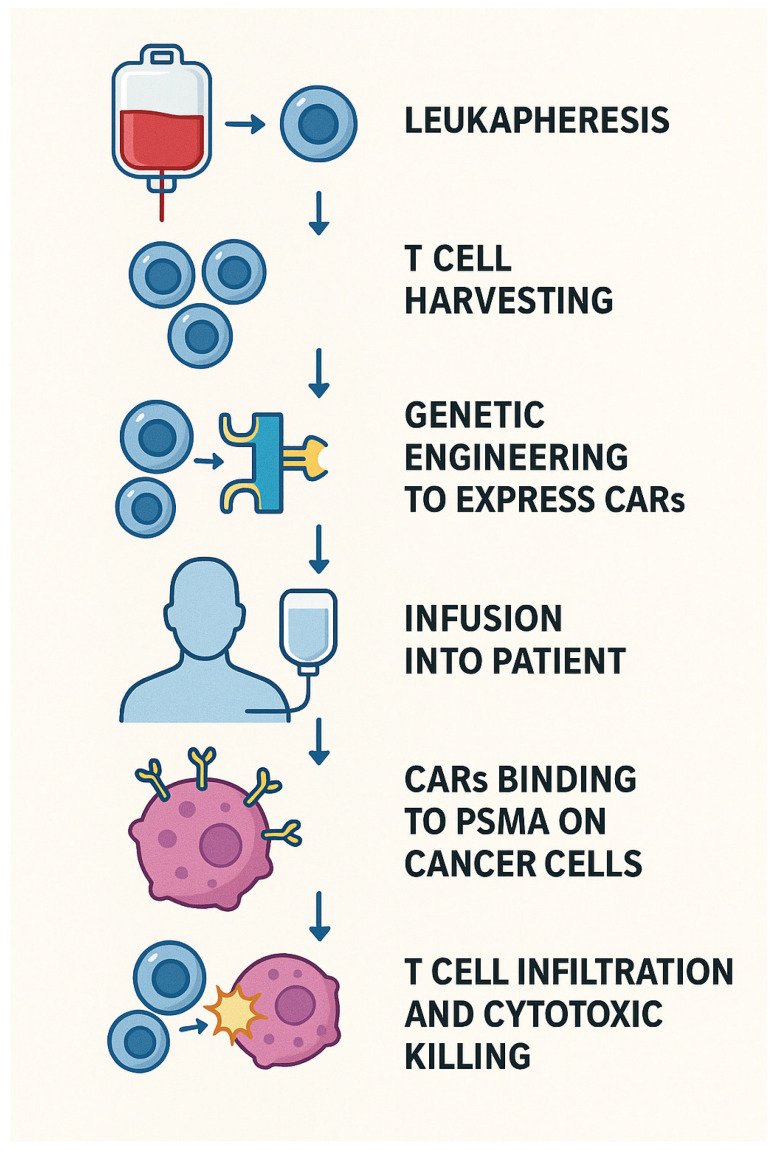
CAR-T MOA.

**Figure 6 biomedicines-13-01837-f006:**
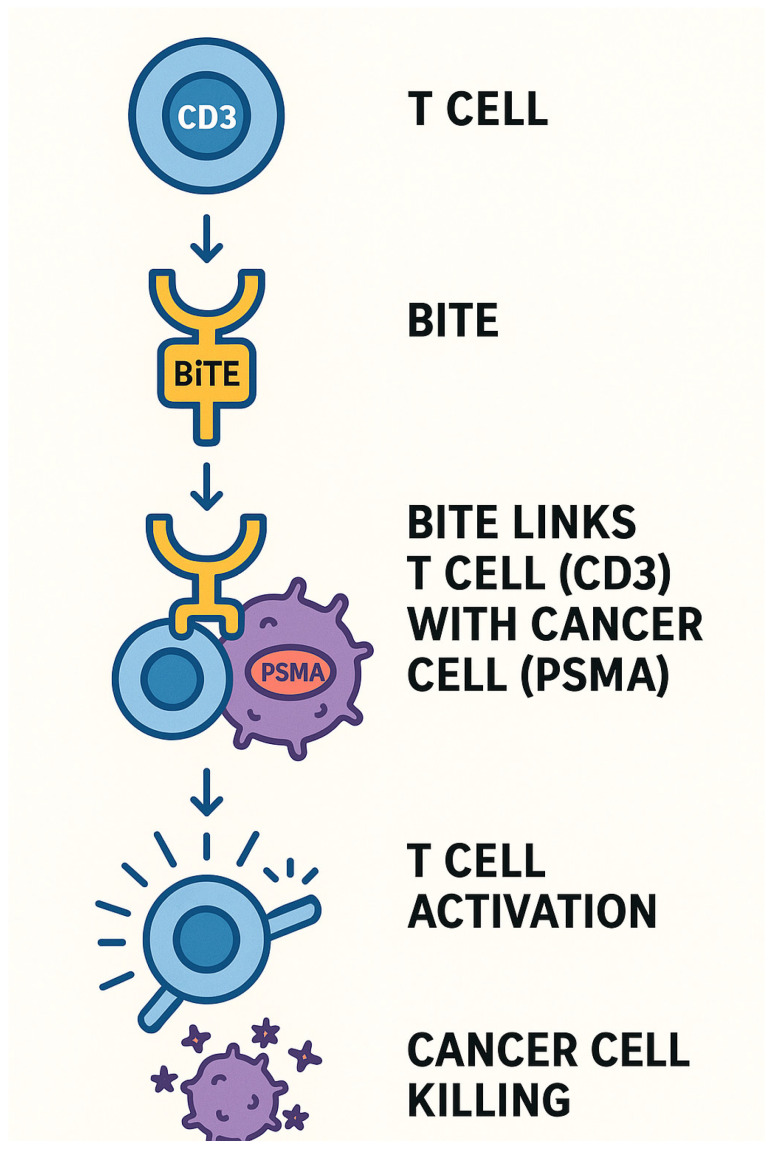
BiTEs MOA.

**Figure 7 biomedicines-13-01837-f007:**
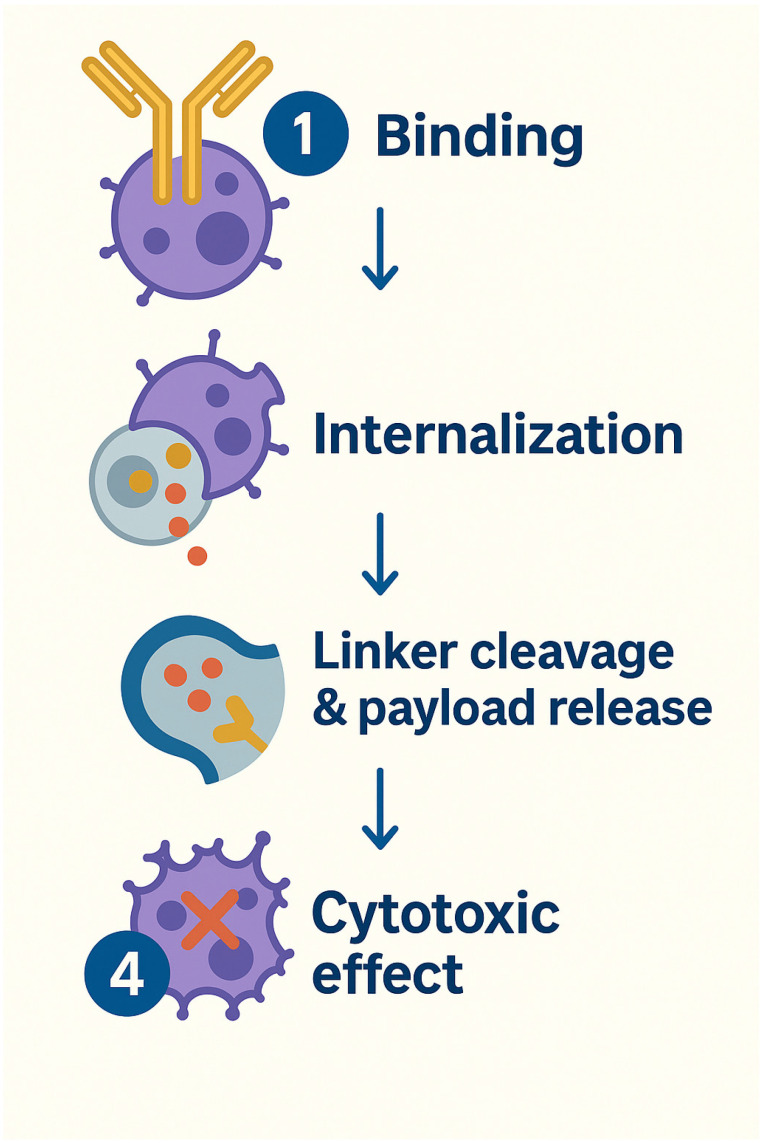
Antibody–Drug Conjugates MOA.

**Figure 8 biomedicines-13-01837-f008:**
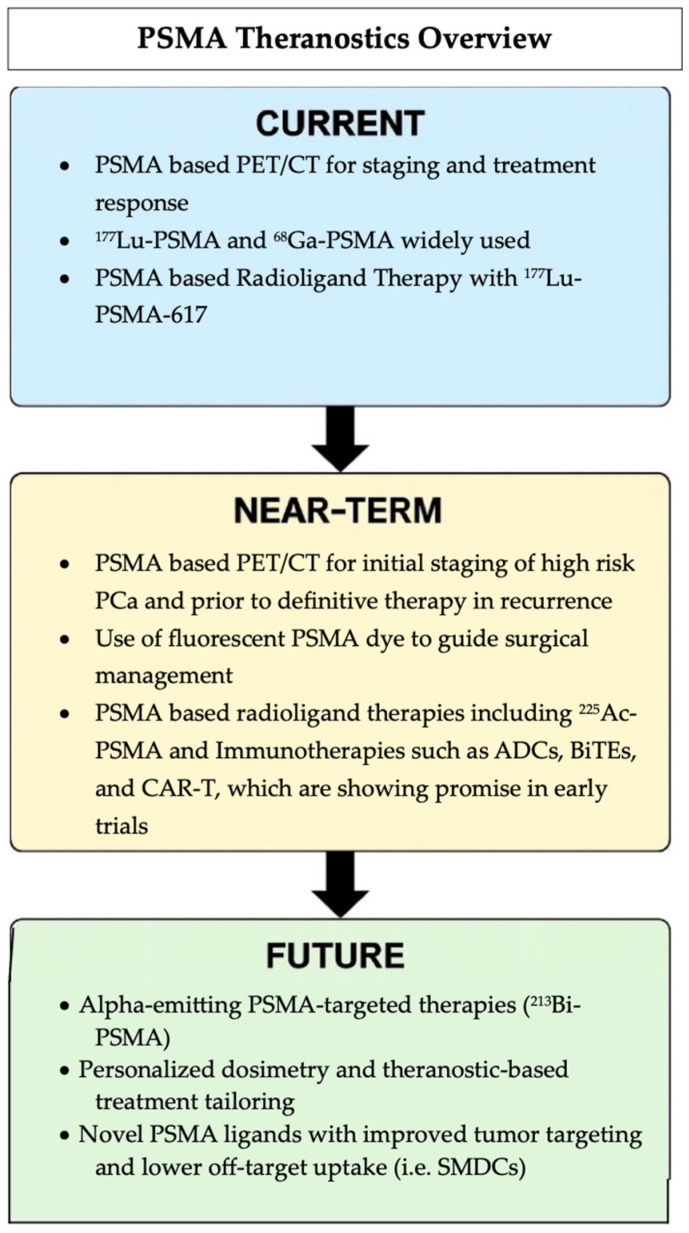
Summary of the evolving PSMA theranostic landscape, from current clinical use to future approaches.

**Table 1 biomedicines-13-01837-t001:** Comparison of FDA-approved PSMA PET tracers based on their use and related trials.

Tracer	Isotope	Brand Name	FDA Approval	Approved Use(s)	Pivotal Trial(s)	Sensitivities and Specificities
^111^In-Capromab	Indium-111	ProstaScint	1996	Initial staging (now obsolete)	Various small pre-PET studies	-
^68^Ga-PSMA-11	Gallium-68	None (academic)	2020	Initial staging, recurrence	UCLA and UCSF Phase 3 Trials (PMID: 33301460)	0.40 (sensitivity), 0.95 (specificity)
^18^F-DCFPyL	Fluorine-18	Pylarify	2021	Initial staging, recurrence	CONDOR (PMID: 33622706), OSPREY	0.40 (sensitivity), 0.98 (specificity)
^18^F-rhPSMA-7.3	Fluorine-18	Posluma	2023	Initial staging, recurrence	SPOTLIGHT (PMID: 37126069), LIGHTHOUSE (PMID: 37414702)	0.60 (sensitivity), 0.96 (specificity)
^68^Ga-Gozetotide *	Gallium-68	Gozellix	2025	Initial staging, recurrence	PSMA-PreRP, PSMA-BCR Trials	0.91 (sensitivity), 0.90 (specificity)

* ^68^Ga-Gozetotide is the same agent as ^68^Ga-PSMA-11, as described above. The primary difference is production.

**Table 2 biomedicines-13-01837-t002:** Ongoing PSMA PET tracer trials, highlighting the possible future direction of PSMA-based therapeutics.

Trial Name	Tracer	Isotope	Phase	Objective
Prospective Multi-Center Study (2023)	PSMA-1007	Fluorine-18	III	Evaluate the diagnostic performance of ^18^F-PSMA-1007 PET/CT in the initial staging of high-risk PCa
Phase III [^18^F] PSMA-1007 for BCR (2021)	PSMA-1007	Fluorine-18	III	Assess lesion detection in patients with biochemical recurrence after prior definitive therapy
PSMAfore (NCT04689828)	PSMA-617	Lutetium- 177	III	Compare ^177^Lu-PSMA-617 vs. ARPI in PSMA+ mCRPC pre-chemotherapy patients
COBRA (NCT05249127)	^18^F-rhPSMA-7.3	Fluorine-18	II	Evaluate the impact of PSMA PET on decision-making for salvage therapy in post-prostatectomy BCR
CLARIFY (NCT06011028)	^18^F-DCFPyL	Fluorine-18	II	Assess PSMA PET-based detection of nodal metastases in patients with suspected recurrence
ProMOTE Study	IR800-IAB2M (PSMA dye)	NIR fluor	I	Investigate the use of fluorescent PSMA dye to guide prostatectomy and improve surgical precision

**Table 3 biomedicines-13-01837-t003:** Various PSMA PET Protocols with associated metrics, strengths, and limitations.

Criteria	Imaging Modality	Key Metric	Strengths	Limitations
RECIST 1.1	CT/MRI	Change in size of target lesions (diameter)	Standardized, widely adopted in solid tumors	Poor sensitivity for bone mets and molecular changes in PCa
RECIP 1.0	PSMA PET/CT	Total PSMA-positive tumor volume + new lesions	Incorporates molecular imaging; high concordance with visual reads	Still undergoing validation; limited availability of software tools
PPP2	PSMA PET/CT	Whole-body PSMA expression + response classification (e.g., PSMA score)	Future-oriented; adaptable framework for prospective trials	Not yet widely adopted; requires PROMISE-compliant data input

**Table 4 biomedicines-13-01837-t004:** Major trials assessing the PSMA radioligand therapy response in mCRPC patients.

Trial	Year	Phase	PCa Stage	No. of Patients	Intervention Regimen	Control Regimen	Primary Endpoint Results
VISION (NCT03511664)	2021	III	mCRPC	831	^177^Lu-PSMA	Standard of Care	PFS:8.7 mo in Intervention vs. 3.4 months in Control
TheraP (NCT03392428)	2019	II	mCRPC	200	^177^Lu-PSMA	Cabazitaxel	PSA response of 66% in Intervention vs. 37% in Control
ENZA-p (NCT04419402)	2024	II	mCRPC	162	^177^Lu-PSMA + Enzalutamide	Enzalutamide	PSA PFS: 13.0 mo in Intervention vs. 7.8 mo in Control
PSMAfore (NCT04689828)	2024	III	mCRPC	468	^177^Lu-PSMA	ARPI	PFS = HR: 0.41 (0.29–0.56)

**Table 5 biomedicines-13-01837-t005:** Upcoming trials assessing novel radioactive isotope efficacy in mCRPC.

Trial Number	Radioactive Novel Isotope	Particle Emission	Phase	Outcomes
VIOLET (NCT05521412)	^161^Tb-PSMA	β	I/II	MTD^1^, AE^2^, SAEs^3^, DLTs^4^, RP2D^5^
NCT03724747	^227^Thorium	α	I	MTD^1^
TheraPb (NCT05720130)	^212^Pb	Β,α (via daughters)	I/II	RP2D^5^

MTD^1^, maximum tolerated dose; AE^2^, adverse events; SAEs^3^, serious adverse events; DLTs^4^, dose limiting toxicities; RP2D^5^, recommended phase 2 dose.

**Table 6 biomedicines-13-01837-t006:** Upcoming trials assessing CAR-Therapyin mCRPC.

Drug	NCT	Phase	Target	Mechanism of Action
CART-PSMA-TGFβRDN	NCT04227275	I	PSMA	Autologous T-cells expressing a PSMA-targeted CAR plus a dominant-negative TGFβ receptor (TGFβRDN), enabling tumor-cell lysis while resisting TGFβ-mediated immunosuppression
PD1-PSMA-CART	NCT04768608	I	PSMA	T-cells engineered via CRISPR/Cas9 to knock out PD-1 and knock in an anti-PSMA CAR at the same locus, providing PSMA-directed cytotoxicity with resistance to PD-1/PD-L1 exhaustion
LIGHT-PSMA-CART	NCT04053062	I	PSMA	Lentivirally transduced PSMA-CAR T-cells co-expressing LIGHT (a TNF-superfamily ligand) to engage HVEM/LTβR, boosting T-cell proliferation, cytokine release, and TME remodeling
P-PSMA-101 CAR-T	NCT04249947	I	PSMA	Autologous T-cells modified with Poseida’s piggyBac transposon to express a PSMA-specific CAR enriched for stem-cell memory phenotypes, enhancing persistence and durable antitumor activity

**Table 7 biomedicines-13-01837-t007:** Summary of the trials exploring the efficacy and safety of BiTes.

Drug	NCT	Phase	Target	CD-3 Affinity	Fc Domain
AMG 340	NCT04740034	I	PSMA	Low	Yes
JNJ-80038114	NCT05441501	I	PSMA	Moderate	Yes
REGN5678	NCT03972657	I	PSMA	Moderate	Yes
REGN4336	NCT05125016	I	PSMA	High	Yes
CC1	NCT04104607	I	PSMA	Moderate	Yes
LAVA-1207	NCT05369000	I/IIa	PSMA	Low	Yes
AMG160	NCT03792841	I	PSMA	Moderate	Yes
CB307	NCT04839991	I	PSMA	NA	No

**Table 8 biomedicines-13-01837-t008:** Antibody–Drug Conjugates (ADC) Summary highlighting the mechanism of action and proposed indication.

ADC	Target	Drug/Payload	Clinical Indication	Development Status
EC1169	PSMA	Tubulysin B hydrazide (TubBH)	mCRPC	Phase I
lutetium (^177^Lu) rosopatamab tetraxetan	FOLH1, PSMA	Radiolabeled with beta-emitting radioisotope Lutetium-177, Lu-177, and conjugated with chelator tetraxetan (DOTA)	mCRPC	Phase III
MLN-2704	PSMA	Maytansine analog drug maytansinoid-1	PCa ^1^	Phase-I/II Discontinued
Pelgifatamab corixetan	FOLH1, PSMA	Chelator corixetan conjugated to an average of 1 lysine	mPCa	Phase I
PSMA-ADC	FOLH1, PSMA	Monomethyl Auristatin E (MMAE)	PCa, Glioblastoma Multiforme (GBM)	Phase I/Phase II
Rosopatamab Tetraxetan	FOLH1, PSMA	Tetraxetan (DOTA), a chelator for yttrium-90, a radioisotope.	PCa	Phase II
ABBV-969	PSMA, STEAP 1	Proprietary topoisomerase-1 (Top1) inhibitor linker-drug format	mCRPC	Phase I
ARX517	PSMA	AS269/pAF-AS269, a proprietary microtubule inhibitor	mCRPC	Preclinical

^1^ According to 2016 Urologic Onc. [84] development was discontinued as a result of dose-dependent peripheral neuropathy (payload).

**Table 9 biomedicines-13-01837-t009:** Summary of reported clinical response rates and key safety signals for PSMA-targeted CAR-T-cells, BiTEs, and ADCs in mCRPC.

Modality	Agent	Response Assessment	Key Safety Signals
CAR-T	PSMA-TGFβRDN	PSA30 ~30.1%	CRS (grade 1–2), ICANS/MAS at higher dose (Grade 5)
	P-PSMA-101	PSA/radiographic responses (unquantified)	Mostly mild CRS, no neurotoxicity
BiTEs	Acapatamab (AMG 160)	PSA50 ~30%, ORR ~7%, PFS ~3.3 months	CRS common (early onset), manageable
	JNJ-081	PSA decline at high dose; no ORR	Mild CRS; generally tolerable
ADCs	PSMA-ADC (MMAE payload)	PSA30 ~30%, PSA50 ~14%, CTC conversion ~75%	Neutropenia, neuropathy, sepsis (dose-related)
	MEDI3726 (PBD dimer)	CRR ~12.1%	TRAE ~90%, Therapy discontinuation due to TRAE ~33%

CRR, composite response rate; ORR, objective response rate; ICANS, immune effector cell-associated neurotoxicity syndrome; CRS, cytokine release syndrome; MAS: macrophage activation syndrome.

## Data Availability

No new data sets were created.

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
