# Peer review of "PSMA-Directed Theranostics in Prostate Cancer"

_biomedicines, 2025, doi:10.3390/biomedicines13081837_

Round 1
Reviewer 1 Report
Comments and Suggestions for Authors
1- Many studies have been published in this field recently. However, I believe more research is necessary to cover all relevant studies. Therefore, I recommend removing older references and replacing them with new literature.
2- Some literature studies in this field were missed (for example, DOI: 10.1002/cmmi.1514).
3- The placement and sequence of some figures are not specified in the main text.
4- Captions for some figures are missing.
5- In my opinion, the authors concentrated on the diagnostics and treatment methods of Lu, but they should also include other nanotheranostics drugs for prostate cancer.
6- The conclusion section lacks any discussion of the benefits or advantages of the study's findings.
Reviewer 2 Report
Comments and Suggestions for Authors
The manuscript in its current form appears poorly polished. It seems like a combination of two manuscripts with a common subject, which results in a very confusing structure, starting from the distribution of chapters and subchapters. Additionally, the year 2024 suggests the use of an outdated template.
The figures are referenced in an unusual way. For example, Figure 1 is not mentioned in the text and lacks a caption.
The manuscript contains two Figure 2 and two Figure 3s.
The separation of chapters, subchapters, and subsections needs to be reviewed.
The abbreviations require revision (what is meant by SMDCs?).
Overall, the manuscript has the potential to be a good review, but at this stage, it appears superficial, confuse and requires significant editing.
Reviewer 3 Report
Comments and Suggestions for Authors
Dear Authors and Editors,
The paper presents a clear and logical structure, taking the reader through PSMA diagnostics and then therapeutics in a stepwise, easy-to-follow manner. This narrative flow is a strength, as it mirrors the way clinicians think about disease management. However, the numbering of subsections occasionally feels awkward and inconsistent, such as “2.2.3.18. F-DCFPyL,” which can confuse readers trying to navigate the text. Adopting a more consistent and intuitive numbering scheme would make it easier to track the discussion.
The depth of coverage is commendable, especially in the sections that summarize major FDA-approved tracers and landmark trials such as VISION and ProPSMA. These summaries are concise and informative. Still, the presentation of data could be improved by moving some of the key sensitivity and specificity numbers from the text into a well-organized table. This would allow readers to quickly compare the clinical performance of the different agents at a glance, enhancing the utility of the review.
The inclusion of figures and tables is a good decision and helps to illustrate complex concepts like mechanisms of action and trial outcomes. However, the captions of these figures tend to be too brief and technical, providing little context to guide the reader. Adding a more descriptive, one-sentence explanation to each figure would improve readability and make them more useful for those who are less familiar with the field.
There are also minor issues with consistency and editing. For example, the Keywords section lists “Prostate Cancer Therapeutics” twice, and the full name “prostate-specific membrane antigen” is repeated unnecessarily in several adjacent paragraphs. Streamlining these repetitions and correcting minor oversights would improve the overall polish of the manuscript.
The balance of content across sections works well overall, with the radioligand therapy section being especially thorough and well-supported by data. In contrast, the sections on immunotherapies and antibody-drug conjugates, while intriguing, are noticeably lighter on clinical data and examples. Including a small table summarizing key response rates and safety signals for these newer modalities would help those sections feel as robust as the discussion of radionuclides.
It is a thoughtful touch that the paper acknowledges issues such as cost and availability of PSMA PET imaging, which adds a real-world perspective. However, the discussion of accessibility could go a step further by addressing how centers in lower-resource settings might adapt these technologies, for example through emerging lower-cost tracers or mobile cyclotron solutions.
The tone of the paper is very academic and comprehensive, which suits its purpose as a specialist review. Even so, weaving in a short illustrative story—perhaps a patient vignette or anecdote showing how PSMA PET or therapy altered a man’s prognosis or treatment plan—would add a human dimension and remind readers of the real-world impact of these advances.
The conclusion does a good job of emphasizing how quickly the PSMA theranostics field is evolving and mentions promising agents on the horizon. Strengthening this section with a simple roadmap figure showing the current standards, near-term innovations, and future directions could leave readers with a clear, memorable vision of where the field is heading.
Overall, this is a timely and well-researched review that succeeds in providing a comprehensive overview of PSMA-based diagnostics and therapies. With some minor improvements to presentation, balance, and narrative warmth, it could become an even more engaging and useful resource for clinicians, researchers, and trainees alike.
Best,
Reviewer
Round 2
Reviewer 2 Report
Comments and Suggestions for Authors
The manuscript has been extensively revised and corrected. Some imperfections still remain. In particular, the legend for Figure 5 and Table 9 are not correctly positioned. Overall, the manuscript is acceptable for publication.